# Recent Strategies to Address Hypoxic Tumor Environments in Photodynamic Therapy

**DOI:** 10.3390/pharmaceutics14091763

**Published:** 2022-08-24

**Authors:** Yuyin Du, Jianhua Han, Feiyang Jin, Yongzhong Du

**Affiliations:** Institute of Pharmaceutics, College of Pharmaceutics Sciences, Zhejiang University, 866 Yu-Hang-Tang Road, Hangzhou 310058, China

**Keywords:** photodynamic therapy, hypoxia, oxygen supply, oxygen

## Abstract

Photodynamic therapy (PDT) has become a promising method of cancer treatment due to its unique properties, such as noninvasiveness and low toxicity. The efficacy of PDT is, however, significantly reduced by the hypoxia tumor environments, because PDT involves the generation of reactive oxygen species (ROS), which requires the great consumption of oxygen. Moreover, the consumption of oxygen caused by PDT would further exacerbate the hypoxia condition, which leads to angiogenesis, invasion of tumors to other parts, and metastasis. Therefore, many research studies have been conducted to design nanoplatforms that can alleviate tumor hypoxia and enhance PDT. Herein, the recent progress on strategies for overcoming tumor hypoxia is reviewed, including the direct transport of oxygen to the tumor site by O_2_ carriers, the in situ generation of oxygen by decomposition of oxygen-containing compounds, reduced O_2_ consumption, as well as the regulation of tumor microenvironments. Limitations and future perspectives of these technologies to improve PDT are also discussed.

## 1. Introduction

The tumor is characterized by the extremely uncontrolled growth of a series of cells, which can spread to other parts of bodies and bring threatening consequences to bodies. The tumor consists of cellular components and the tumor microenvironment (TME). The TME, which is composed of extracellular matrix, stromal cells, and immune cells, has been found to play a significant role in tumor progression [1]. Hypoxia, which arises when there is an imbalance between oxygen supply and consumption and is characterized by the O_2_ pressure < 10 mm Hg, is one of the important factors of tumor microenvironments [2,3]. Hypoxia-inducible factors (HIF) are transcription factors of downstream hypoxia-inducible genes, composed of oxygen-labile subunit HIF-α and a constitutively expressed HIF-β [4]. Among them, HIF-1α regulates the expression of genes that are involved in tumor progression, aggressiveness, and metastasis [5]. Under the hypoxic state, HIF-1α is activated and highly expressed. As a result, the hypoxic state in tumor leads to the aggressive phenotype, induces tumor progression, promotes tumor spreading, and causes angiogenesis [6,7,8]. Additionally, a great deal of evidence has also proved that hypoxia induces tumor resistance to various therapies via decreased drug-induced senescence [9,10]. Therefore, tumor cells exposed to hypoxia are much more aggressive and resistant. 

Photodynamic therapy (PDT) has progressively played solid roles in cancer treatment owing to its inherent merits, such as low toxicity, minor side effects, and accurate target treatment without damage to adjacent normal tissues [11,12,13,14]. Fundamentally, PDT is composed of three components: light, a photosensitizer (PS), and oxygen [15]. Under laser irradiation, the PS is activated to generate reactive oxygen species (ROS) through the type I mechanism based on radical reaction or through the type II mechanism, where the energy is directly transferred to triplet ground state oxygen (^3^O_2_) to generate reactive singlet oxygen (^1^O_2_) [16]. The generated ROS, which is cytotoxic, could further induce cytotoxicity, cell damage, or death. While PDT possesses various advantages, it still has several limitations, such as the need for ideal PS and tumor hypoxia environments [17,18]. Until now, much research has been carried out to develop the ideal PS for phototherapies, such as gold nanoparticles (AuNPs) [19,20,21], chlorin e6 (Ce6) [22], and indocyanine green (ICG) [23,24]. Among the limitations of PDT, the hypoxia environment is one of the most significant factors. Since the PDT process involves especially high consumption of oxygen, the hypoxic character of tumor dramatically reduces PDT efficacy, and the hypoxia also deteriorates simultaneously. Consequently, the increased tumor hypoxia induces cancer progression and metastasis, leading to the higher risk of PDT resistance [25]. 

In spite of challenges in alleviating tumor hypoxia, significant efforts have been made to address the limitation of PDT caused by hypoxia in recent years [26,27,28]. These strategies are mainly divided into several directions, including (1) the direct transport of oxygen by O_2_ carriers, (2) in situ O_2_ generation by catalase or O_2_ self-supply materials, (3) a decrease in oxygen consumption, and (4) other strategies, such as increasing blood flow and downregulating HIF-1 (Figure 1). Here, we review and discuss recent research designs to deal with tumor hypoxia in PDT from these directions. Finally, their limitations and future perspectives are also discussed in this review. 

## 2. Direct Transport of Exogenous Oxygen to Tumors

To alleviate the tumor hypoxia in PDT, the transportation of exogenous oxygen to the tumor is a straight way. Hemoglobin, perfluorocarbon groups, and metal–organic frameworks are common O_2_ carriers used to deliver O_2_ to the tumor cells. These O_2_ carrier-contained nanoplatforms are summarized in Table 1.
pharmaceutics-14-01763-t001_Table 1Table 1Summary of recent studies on direct transport of oxygen by O_2_ carriers.O_2_-CarrierNanosystemsPSStrategyReferenceHemoglobinGd@Hb^Ce6-PEG^Ce6Hb increased tumor oxygenation and MR imaging-guided enhanced PDT.[29]
SPN-Hb@RBCMSPNsProlonged circulation time and accumulation of nanoparticles in tumor by RBCM enhanced PDT treatment efficacies.[30]
HbTcMsTCPPHbTcMs could produce singlet oxygen and kill 4T1 cells under irradiation.[31]
Hb/ICG loaded artificial red cellsICGHb supplied sufficient oxygen for PDT and self-monitoring PDT was performed through FL/PA dual-modal imaging.[32]PerfluorocarbonPFP, ICG and BPD contained nanodropletsICGThe systems could produce photoacoustic contrast, deliver oxygen, and improve PA-guided PDT efficacy.[33]
CCm-HSA-ICG-PFTBAICGThe nanoprobe was targeted to tumor tissues, mitigated hypoxia, and enhanced the PDT efficacy.[34]
O_2_@PFOB@PGL NPsPorphyrinThe nanoparticles alleviated hypoxia under fluorescence/CT imaging guidance.[35]
Ce6-PFOC-PEI-MCe6PFOC provided high oxygen-carrying ability, enhancing higher PDT efficacy.[36]MOFsCuTz-1-O_2_@F127CuTz-1CuTz-1@F127 could then deliver O_2_ generated by PS to tumor cells, alleviating hypoxia.[37]
O_2_@UiO-66@ICG@RBCICGUiO-66 MOFs could store O_2_ and release it upon light irradiation, relieving hypoxia.[38]

### 2.1. O_2_ Carriers Based on Hemoglobin Groups

In human bodies, red blood cells are fundamental tools for oxygen transportation under the help of hemoglobin. The hemoglobin molecule consists of four heme groups where an iron ion is ligated at the center of a porphyrin ring. Each of the four heme groups can strongly bind to one molecular O_2_ molecule, which can be carried from the lungs to other tissues by the hemoglobin inside red blood cells in human bodies, exhibiting the excellent O_2_ capacity of Hb. To mitigate the hypoxia environment in tumor, many strategies have been made to increase the endogenous oxygen content by utilizing the hemoglobin-based oxygen carriers. Although hemoglobin is known for its excellent oxygen loading and releasing capability, its relatively short lifetime and instability cause side effects to patients [39]. Thus, to address its shortcomings, Hb has been modified by physical encapsulation or chemical conjugation [40,41,42].

For instance, PEGylation of Hb can improve the biocompatibility, enhance the stability, and prolong the half-life of hemoglobin during the circulation. Shi et al. designed the nanostructures named Gd@Hb^Ce6-PEG^ with PDT function through protein-mediated biomimetic synthesis, which is an eco-friendly technology used in the biomedical field and is a simpler process compared with complexed chemical synthesis. Firstly, Hb was PEGylated, followed by the cooperation of Gd nanoparticles, photosensitizer Ce6, and oxygen. Significantly, Hb in this system not only showed excellent biocompatibility and intrinsic oxygen-carrying capability, but also provided the reaction region for the biomimetic chemistry of Gd nanoparticles. The in vitro experiments in this study showed that the viability of 4T1 cells incubated with Gd@HbCe6-PEG and oxygenated Gd@HbCe6-PEG and laser irradiation decreased to 34% and 18%, respectively, indicating efficient killing of cancer cells. The results of PA imaging and HIF-1α protein expression analysis demonstrated that this system could efficiently deliver O_2_ and alleviate tumor hypoxia [29].

Since hemoglobin carries O_2_ inside the red blood cells (RBCs), RBCs have also been widely used as drug carriers. Nanoscale RBCs or red blood cell membrane (RBCM)-modified nanoparticles (NPs) are an effective way to ensure tumor permeability or internalization [43]. For example, Ding et al. constructed a red blood cell biomimetic nanovesicle (SPN-Hb@RBCM), as shown in Figure 2. In this system, the oxygen-supplying hemoglobin was linked with semiconducting polymer nanoparticles (SPN), which was a theranostic agent for near-infrared (NIR)-II fluorescence imaging and singlet oxygen (^1^O_2_) production for PDT. The resulting SPN-Hb was further coated with RBCM to form SPN-Hb@RBCM. Notably, this RBCM coated system promoted longer circulation time and nanoparticle accumulation in vivo, allowing for enhanced treatment efficacies. The in vitro experiment result showed that the overall oxygen-carrying capacity of SPN-Hb@RBCM nanoparticles was determined and calculated to be 0.142 μg of O_2_ per 1 mg NPs. The cell-killing efficacy was 90.3% in the SPN-oxy-Hb@RBCM group and was only 45.1% in the SPN@RBCM groups, showing that the SPN-oxy-Hb@RBCM with oxygen self-supply ability could retain its great PDT efficiency under hypoxia environments. Although this strategy could improve hypoxia to a certain extent, the oxygen loading content of the nanomaterial was limited, which may restrict the practical application [30]. 

### 2.2. O_2_ Carriers Based on Perfluorocarbon Groups

Since hemoglobin has several limitations, such as susceptible conformation changes and a short circulation time, perfluorocarbon (PFC) is an alternative oxygen carrier. Perfluorocarbons (C_n_F_2n+2_), show extremely strong O_2_ affinity due to their high electronegative fluorine, biocompatibility, and high stability, thus representing good oxygen carriers [44,45]. Significantly, PFCs can dissolve an enormously large amount of oxygen compared with water. For example, the molecular ratio of dissolved O_2_ is 5:1 in perfluorodecalin (PFD), but 1:200 in water, which means PFD shows the 1000 times increased molecular solubility compared with water [46]. In addition, it is also reported that the lifetime of singlet oxygen in PFC is longer than in the intracellular environment or in water [47]. Hence, in these years, numerous research studies have been carried out to deal with tumor hypoxia and improve PDT efficacy by introducing PFC-based O_2_ carriers. 

Xavierselvan et al. designed nanodroplets that were composed of O_2_-saturated perfluoropentane, indocyanine green (ICG) that has peak light absorption in the NIR range, as well as benzoporphyrin derivative (BPD). This research showed the 9.1-fold increase in oxygen after the administration and irradiation of nanodroplets, proving the outstanding oxygen supply capability to the hypoxia site. Both in vitro and in vivo murine tumor model studies conducted in this research suggested that these PFP containing nanodroplets loaded with photosensitizer have shown to be able to enhance photoacoustic contrast, deliver oxygen, combat hypoxia, and increase PDT efficacy [33].

Although ICG can be utilized for deep tissue penetration, fluorescence quenching may occur due to the aggregation and short time of blood circulation. To solve this obstacle, Fang et al. has constructed an oxygen delivery nanoprobe, HSA-ICG-doped perfluorotributylamine, coated by the cancer cell membrane (CCm–HSA–ICG–PFTBA), to target and alleviate the hypoxia in tumor tissues, as shown in Figure 3a. Human serum albumin (HSA) was used to stabilize the photosensitizer ICG and prolong the circulation time. The coating with cancer cell membrane could stabilize nanoparticles and had the ability to perform homologous targeting. The in vivo ^18^F-fluoromisonidazole (^18^F-FMISO) positron emission tomography/computed tomography (PET/CT) imaging was performed in this research through the scheme in Figure 3c. The imaging and quantitative analysis results (Figure 3b,e) showed that the value of SUVmax significantly decreased after the treatment of CCm–HSA–ICG–PFTBA. Moreover, the ex vivo immunofluorescence staining imaging results shown in Figure 3d,f indicated that the hypoxia area was prominently reduced from 87.4% to 8.3% after the injection of the nanosystem. These results indicated the high oxygen level delivers the capability of PFTBA, relieving tumor hypoxia and improving PDT efficacy [34].

### 2.3. O_2_ Carriers Based on Metal-Organic Frameworks

Metal–organic frameworks (MOFs) are composed of metal ions coordinating with organic linkers via coordination bonds. MOFs have been widely investigated in the application of sensors, catalysis, and storage due to their porous network, which is capable of trapping and loading diverse molecules and compounds via covalent or noncovalent interactions [48,49]. In recent years, MOFs have been applied in cancer therapy and showed various advantages, such as increased drug loading capability due to the large surface areas and high porosities, as well as high biodegradability because of the weak coordination bonds [50]. By changing the pore geometry and size of MOFs, MOFs which possess oxygen storage ability have also been synthesized [51]. Therefore, MOF-based O_2_ carriers have aroused attractive attention to supply sufficient O_2_ to tumors, overcoming hypoxia.

The Cu-based MOFs possess high O_2_ storage capability owing to the coordinatively unsaturated Cu sites [51]. Therefore, Cai et al. designed an O_2_-loaded CuTz-1@F127 MOF therapeutic system (CuTz-1-O_2_@F127), which is biocompatible and biodegradable. As a light-activated photosensitizer, the CuTz-1 MOF could generate hydroxyl radicals and O_2_ in the presence of H_2_O_2_ through type I PDT. The in vitro study revealed that at standard atmospheric pressure and room temperature, the oxygen-carrying capacity was adsorbed up to 400 μmol O_2_ g^−1^ of CuTz-1-O_2_@F127 (12.8 μg of O_2_ per 1 mg,), which is much higher than the previously mentioned SPN-Hb@RBCM NPs. The CuTz-1@F127 could then deliver O_2_ into cancer cells, thus alleviating intracellular hypoxia. At the same time, the CuTz-1@127 could absorb overexpressed intracellular glutathione (GSH), which would significantly reduce the cytotoxicity of ROS during the treatment of cancers. The cell study showed that the cell viability of the group treated with CuTz-1@127 reduced to approximately 20%, while the cell viability was 100% for the control group. Thus, in this research, the efficiency of PDT was greatly enhanced both through the increased O_2_ supply and reduced level of GSH [37].

In addition, a zirconium (IV)-based MOF, UiO-66, has been widely used for O_2_ storage due to its tunable pore sizes and high specific surface areas [51]. Inspired by this, Gao et al. fabricated a biomimetic O_2_-evolving PDT nanoplatform (O_2_@UiO-66@ICG@RBC), where the ICG was anchored on the surface of the UiO-66 and the NPs were encapsulated by RBC membranes. Under the 808 nm laser irradiation, ICG could generate singlet oxygen through energy transfer and thus degrade RBC membranes. The oxygen stored by UiO-66 would be released by the heat converted from NIR light, overcoming hypoxia. In this research, PA imaging was used to evaluate the ability of O_2_@UiO-66@ICG@RBC to induce oxygenation by measuring oxygenated hemoglobin (HbO_2_). The results showed that the signal intensity of HbO_2_ was significantly increased in the group treated with O_2_@UiO-66@ICG@RBC, while the signal intensity was not obvious for the groups treated with UiO-66@ICG@RBC or saline. The in vivo PDT also showed efficient tumor growth inhibition owing to the excellent tumor accumulation and oxygen capacity of O_2_@UiO-66@ICG [38].

Despite the fact that a great number of excellent research has been carried out successfully to develop the NPs to carry oxygen, the ratio of the amount of PSs to the amount of transported O_2_ is not given in these studies. This value should be studied and included in future O_2_-carrying PDT research, as it is an essential parameter to suggest the efficiency of the PS in transporting O_2_ in order to alleviate the hypoxia. In addition, the biocompatibility and biosafety of MOFs should be carefully assessed for clinical application, as some metal ions may have harmful effects on human bodies.

## 3. In Situ Oxygen Generation

Compared with O_2_ carriers, the O_2_ generation strategy can avoid the leakage of O_2_ during transport, showing more potential to alleviate hypoxia. Through decomposition of oxygenated compounds such as H_2_O_2_, CaO_2,_ or water, the oxygen can be generated in situ to supply O_2_ to tumor tissues. Since H_2_O_2_ is reported to be present in tumor tissues, H_2_O_2_ is the most common oxygen source, as it can decompose to generate O_2_. Additionally, some metal oxides such as CaO_2_ and Au_2_O_3_ can produce oxygen via decomposition with catalysts. H_2_O is also a common oxygen source, as water is abundant in many organisms. C_3_N_4_ or photosynthesis can turn water into O_2_. Here, we summarize and review various nanoplatforms with these oxygen generation compounds that have been constructed to alleviate hypoxia and enhance PDT in recent years (Table 2).
pharmaceutics-14-01763-t002_Table 2Table 2Summary of recent studies based on in situ O_2_ generation strategies.Decomposition CompoundsNanosystemsPSStrategyReferenceH_2_O_2_HCINPsIR820CAT catalyzed the decomposition of H_2_O_2_ to generate O_2_.[52]
Cat@PDSDiDCAT catalyzed the decomposition of H_2_O_2_ to generate O_2_.[53]
PEG-Ce6-PEI@PBCe6PB catalyzed the decomposition of H_2_O_2_ to generate O_2_.[54]
MnO_2_@TPP-PEGTPPMnO_2_ catalyzed the decomposition of H_2_O_2_ to generate O_2_ and ∙OH.[55]
ICG@PEI—PBA—HA/CeO_2_ICGCeO_2_ catalyzed the decomposition of H_2_O_2_ to generate O_2_ to mitigate hypoxia.[56]
Co/ZIF-8/ICG/PtICGPt catalyzed the decomposition H_2_O_2_ to enhance PDT and CDT.[57]
Pt-ICG@PDA NPsICGPt catalyzed the decomposition H_2_O_2_ to enhance PTT/PDT.[58]
ICG-PtMGs@HGdICGPt catalyzed the decomposition H_2_O_2_ to generate O_2_ to enhance PDT/PTT effect.[59]CaO_2_LipoMB/CaO_2_MBCaO_2_ would react with water to generate O_2_ and alleviate hypoxia.[60]
CMP-MBMBCaO_2_ would react with water to form H_2_O_2_ which is further catalyzed by MnO_2_.[61]Au_2_O_3_UCSiAuO-2Ce6Au_2_O_3_ was decomposed to O_2_ through FRET effect from UCNPs.[62]H_2_OCDots/TiO_2_ NTsCDots/TiO_2_CDots/TiO_2_ generated O_2_ from water splitting and ROS.[63]
Ru-g-C_3_N_4_C_3_N_4_The nanosheet could decompose water and H_2_O_2_ to generate O_2_ and ROS.[64]
C_5_N_2_ NPsC_5_N_2_The NPs simultaneously split water to generate O_2_ and ROS under NIR light irradiation.[65]
UCTM NPsChlorophylPS-I and PS-II could split water into O_2_ and support the ROS formation.[66]

### 3.1. Decomposition of Endogenous H_2_O_2_ by Catalase and Nanozymes

Studies have shown that the amount of hydrogen peroxide, H_2_O_2_, is considerably higher in cancer cells compared with normal cells, which is related to DNA damage, abnormal cell proliferation, metastasis, and angiogenesis [67,68]. Since H_2_O_2_ can be decomposed to form O_2_ and H_2_O by catalase (CAT) or nanozymes, many nanoplatforms are developed to utilize H_2_O_2_ as oxygen generator to address tumor hypoxia.

Catalase is an antioxidant enzyme which contains iron porphyrin as its auxiliary group and can catalyze the decomposition of H_2_O_2_ into O_2_ and H_2_O. While biological catalase can catalyze H_2_O_2_ and generate oxygen to enhance the local oxygen concentration of tumors, free CAT is difficult to accumulate inside tumors due to its low stability [69]. Therefore, to increase its stability, CAT is encapsulated inside nanocarriers to separate them from molecules such as proteases. For example, nanoparticles HA-PLGA-CAT-IR820 (HCINPs) were developed by Hou et al. In this system, CAT, and the PS IR820 were incorporated into poly(lactic-*co*-glycolic acid) (PLGA), a polymer with excellent biodegradability and biocompatibility. Hyaluronic acid (HA) was modified on the surface of nanoparticles for its tumor-targeting ability (Figure 4a). The researcher used ultrasound imaging and optical microscopy to confirm the oxygen generation, as shown in Figure 4b,c. The results showed that CAT could produce oxygen by catalyzing H_2_O_2_ to alleviate the tumor hypoxic state. The oxygen was then converted to single oxygen via irradiation with IR820, resulting in the PDT enhancement [52]. Similarly, Yin et al. used a PEGylated phospholipids membrane to protect catalase from protease degradation. They embedded the photodynamic agent of DiD and cytotoxic soravtansine into the DSPE-PEG, and then encapsulated with catalase nanoparticles to form Cat@PDS. The treatment of this system under the irradiation could significantly increase the oxygen amount, reduce the expression of HIF-1α, and generate ROS. This research also showed that the Cat@PDS upon laser irradiation resulted in a 97.2% inhibition of lung metastasis, representing an effective chemo–photodynamic therapy [53]. 

Prussian blue (PB) possesses catalase-like activity, providing the capability of catalyzing the decomposition of H_2_O_2_ [70]. The nanoplatform PEG-Ce6-PEI@PB was developed by Wang et al. for enhanced PDT, where the polyethylenimine (PEI) was used as the cationic polymer. The PB in PEG-Ce6-PEI@PB nanoparticles had catalase-like activity to catalyze H_2_O_2_ for O_2_ generation to amplify the following PDT. Both the in vitro and the intracellular study showed the catalytic ability of PEG-Ce6-PEI@PB to produce O_2_ [54]. 

Another way to catalyze the decomposition of H_2_O_2_ to O_2_ is to use manganese dioxide (MnO_2_) nanoparticles [71]. Chen et al. developed an amphiphilic polymer by using photosensitizer TPP and epoxy-ethyl-terminated oligomeric ethylene glycol, followed by self-assembly with MnO_2_. The resulted nano-enzyme, MnO_2_@TPP-PEG, aggregated at the tumor site via a passive targeting effect and disintegrated by laser irradiation to release MnO_2_. The released MnO_2_ could further catalyze the decomposition of H_2_O_2_ to produce O_2_, thus alleviating tumor hypoxia to provide the enhanced potential for PDT. Except for O_2_, cytotoxic ˙OH could also be generated by the decomposition of H_2_O_2_ to achieve CDT for the enhanced killing of tumor cells in this study [55].

Except for MnO_2_, cerium oxide (CeO_2_) is also commonly used to catalyze the decomposition of H_2_O_2_ for PDT. A CeO_2_-based nanozyme, ICG@PEI − PBA − HA/CeO_2_, was synthesized by Zeng et al., as shown in Figure 5a. The HA was able to target the ICG-loaded nanovesicles to the HA receptors expressed on the tumor site. The pH cleavage of phenylboronic acid (PBA) triggered the release of CeO_2_ and ICG at the tumor site. By catalyzing the generation of oxygen from H_2_O_2_, CeO_2_ could regulate the hypoxia tumor microenvironment. In this system, CeO_2_ nanozymes were regenerable CAT-like nanozymes, in which Ce^4+^ was converted to Ce^3+^ when exposed to H_2_O_2_ as shown in Figure 5b. Additionally, the released ICG could induce apoptosis through laser-mediated PDT and photothermal therapy (PTT) [56].

In addition, since the metal platinum (Pt) is one of the catalysts for the decomposition of H_2_O_2_, Pt NPs were developed to boost the catalysis [72]. Jiang et al. designed a nanoplatform Co/ZIF-8/ICG/Pt (CZIP), which was based on zeolitic imidazolate framework 8 (ZIF-8) that could degrade in the acidic environment. Pt nanoparticles were coated on the surface of Co^2+^-doped ZIF-8 to catalyze H_2_O_2_ to produce O_2_. In this study, CZIP nanoplatforms not only improved PDT efficacy through the generation of O_2_ by Pt NPs, but also enhanced CDT through the production of ^•^OH through the release of Co^2+^ [57]. While Pt NPs showed excellent catalytic ability to decompose H_2_O_2_, the size of Pt NPs is, however, usually less than 10 nm, which is very small and easily cleared by the urinary system. Therefore, to address this problem, Cao et al. developed a PTT agent polydopamine (PDA) nanocarrier to load both Pt NPs and photosensitizer ICG. (Pt-ICG@PDA NPs). The conjugation of Pt NPs with PDA efficiently relieved the hypoxia environment. Typically, in vivo study showed that the tumor size ratio (*V/V_o_*) of tumors treated with PDA was 6.67 due to the PTT effect of PDA, and that of tumors treated with ICG@PDA was 3.37 with the combined PTT and PDT effect. Remarkably, the *V/V_0_* value of tumors treated with Pt-ICG@PDA was almost zero, showing the excellent ability of Pt to provide more O_2_ for enhanced PDT [58].

### 3.2. Decomposition of O_2_ Self-Sufficient Metal Oxides

The amount of endogenous H_2_O_2_ in the tumor is limited to 10–50 μM, which is not enough for the in situ generation of oxygen [73,74]. Therefore, other materials that can also release O_2_ in tumors, including calcium peroxide (CaO_2_) and gold oxide (Au_2_O_3_), are alternative strategies. CaO_2_ was reported to be a self-sufficient material that reacts with H_2_O and produces O_2_ [75]. The O_2_ self-sufficient liposome nanoplatform, LipoMB/CaO_2_, was developed by encapsulation of photosensitizer methylene blue (MB) and PEG-stabilized CaO_2_ into the aqueous cavity and the hydrophobic layer of liposome, respectively. With the light irradiation, the activation of ^1^O_2_ broke down the liposomes. The liposome breakdown increased the contact of CaO_2_ to H_2_O, accelerating the speed of O_2_ generation. The results of apoptosis and necrosis analysis suggested that about 20% and 38% of cell death were shown in LipoMB/CaO_2_ groups at single and dual-stage light treatments, which was much higher than LipoMB groups (0.8% and 0.7), indicating that LipoMB/CaO_2_ could alleviate the hypoxia and improve the efficiency of PDT [60]. Notably, CaO_2_ reacts with water to produce H_2_O_2_ and Ca(OH)_2_ (Equation (1)). Thus, a novel nanosystem CaO_2_/MnO_2_@PDA-MB was designed recently, which contained a MnO_2_ nanosheet and CaO_2_ nanoparticles. In this system, MnO_2_ would catalyze and generate O_2_ via decomposition of H_2_O_2_ that was produced by the reaction of CaO_2_ and water [Equation (2)], overcoming the defect of tumor hypoxia [61].
(1)$${\mathrm{CaO}}_{2}+2H_{2}O\to H_{2}O_{2}+\mathrm{Ca}{\left(\mathrm{OH}\right)}_{2}$$
(2)$${\mathrm{MnO}}_{2}+H_{2}O_{2}+2H^{+}\to {\mathrm{Mn}}^{2+}+2H_{2}O+{O_{2}}_{\;}$$

Apart from CaO_2_, Au_2_O_3_ could also be decomposed to produce O_2_ under NIR irradiation through the upconversion and fluorescence resonance energy transfer (FRET) effect. Niu et al. has designed a nanodrug carrier named UCNPs@SiO_2_-Au_2_O_3_-Ce6-RGD-PEG, which was composed of upconversion nanoparticles (UCNPs) that could react with a PS under NIR irradiation. Through the FRET effect, not only could the UCNPs provide energy to activate the photosensitizer Ce6, but also the Au_2_O_3_ could decompose to Au and O_2_. Then, Ce6 could activate the generated O_2_ to produce ^1^O_2_, enhancing PDT in the hypoxia environment. The in vivo evaluation of antitumor properties showed that the mice in the groups treated with UCNPs@SiO_2_-2 + NIR and UCSiAuO-2 + NIR groups had 84% and 95% survival rates, while other groups showed lower than 40% survival rates, obviously indicating that the treatment of the UCSiAuO-2 + NIR could improve the therapeutic effect [62].

### 3.3. Decomposition of Water

Since water is the most abundant substance in living organisms, water can produce an unlimited amount of O_2_ compared with other O_2_-generating materials. The water splitting process usually requires light, catalysts, and water [76,77]. When a photon absorbs a sufficiently short wavelength that exceeds or is equal to the bandgap of the catalyst, electrons can be activated from the conduction band to the valence band, which leads the separated charges to undergo the reduction and oxidation process with nearby water. The water-splitting equation can be generally summarized as Equations (3) and (4).
(3)$$2H_{2}O\to O_{2}+4H^{+}+4e^{-}$$
(4)$$\;4H^{+}+4e^{-}\to 2H_{2}\;$$

In recent years, carbon nitride (C_3_N_4_) has been designed to mitigate hypoxia in tumors because it is a novel biocompatible material that can generate oxygen continuously by synergistically catalyzing the decomposition of H_2_O and H_2_O_2_. Wei et al. designed a graphitic-phase carbon nitride nanosheet coordinated with [Ru(bpy)_2_]^2+^ (Ru-g-C_3_N_4_). The coordination with the ruthenium complex meant the nanosheet had better biocompatibility, higher stability, and water solubility. Moreover, Ru-g-C_3_N_4_ has a narrower bandgap and a shifted valence band compared to g-C_3_N_4_, enabling activation by visible light. The results showed this Ru-g-C_3_N_4_ system could catalyze the decomposition of water or H_2_O_2_ to O_2_ and multiple cytotoxic ROS under visible light exposure, which ameliorated hypoxia and enhanced the PDT [64]. Additionally, an organic semiconductor composed of C_5_N_2_ nanoparticles was also developed by Chen et al. to achieve effective PDT. Importantly, C_5_N_2_ NPs possessed a low valence band position, providing adequate oxidation force for photocatalytic water splitting O_2_ evolution. The O_2_ generation capability was confirmed by the experiment’s result, which showed that a sufficient amount of O_2_ was generated under light irradiation, while no O_2_ was generated under dark environments. This research indicated that C_5_N_2_ NPs could target the cell nucleus and possessed strong photooxidation capacity which could split water to simultaneously generate O_2_ and the singlet oxygen under 650 nm laser irradiation. The generated ROS could achieve enhanced PDT as well as direct DNA damage to further improve the anticancer effect [65]. 

In addition, metal oxide TiO_2_ has attracted much attention as a water-splitting material. TiO_2_ has been evidenced to have low toxicity and high photostability, and can provide a large interior and outer surface area to allow an increase in the light absorption intensity and further modification [78,79,80]. Hence, carbon-nanodot-decorated TiO_2_ nanotubes (CDots/TiO_2_ NTs) were developed for PDT treatment via water-splitting processes, where H_2_O was decomposed to H_2_O_2_ and H_2_O_2_ was further decomposed to water and O_2_, as shown in Equations (5) and (6). Through the upconversion process, the CDots/TiO_2_ produced abundant singlet oxygen to induce apoptosis or necrosis. Additionally, CDots/TiO_2_ could generate a large amount of oxygen through continuous water splitting to improve PDT efficacy [63].
(5)$$2H_{2}O\to H_{2}O_{2}+H_{2}$$
(6)$$2H_{2}O_{2}\to 2H_{2}O\;+O_{2}\;$$

Moreover, the photosynthesis of plants and bacteria also involves the water splitting process, in which the absorption of solar energy converts water into oxygen via thylakoids. Therefore, many researchers have been inspired by this and have developed strategies based on photosynthesis to generate O_2_. Thylakoid membranes in chloroplasts contain a Z-scheme system, which is the location of photosystem I (PS-I) and photosystem II (PS-II). Both PS-I and PS-II contain chlorophyl, which is a photosensitizer that can be activated upon visible light irradiation to generate ROS and O_2_. Nevertheless, PS-I and PS-II can only be captivated by red light, meaning that it cannot penetrate tissue deeply compared with NIR light. To overcome this limitation, Cheng et al. developed upconversion nanoparticles decorated with thylakoid membranes (UCTM NPs), as shown in Figure 6. Upon 980 nm laser irradiation, UC NPs could emit red light, which could activate both PS-I and PS-II to simultaneously generate ^1^O_2_ and O_2_ to enhance PDT for hypoxic tumor treatment. The in vivo O_2_ generation study of UCTM NPs demonstrated the simultaneous ^1^O_2_ and O_2_ production ability. The in vitro PDT efficacy assessments showed that with laser irradiation, UCTM NPs could cause dramatic cell death (higher than 80%), whereas control groups did not induce obvious cell death. The excellent therapeutic effect against the hypoxic tumor was also confirmed with in vivo study [66].

Even though the generation of O_2_ endogenously can advance PDT efficacy, there are still many aspects that need to be carefully addressed by future research. For instance, the amount of endogenous H_2_O_2_ is limited, and the delivery of exogeneous H_2_O_2_ is still difficult. The biosafety and degradation of metal ions, including Au_2_O_3_, should also be assessed.

## 4. Decrease in O_2_ Consumption

The level of tumor hypoxia is related to the balance between the rate of O_2_ supply through the vasculature and the rate of O_2_ consumption in tumor tissues. Therefore, except for increasing the concentration of oxygen inside tumors, reducing the amount of oxygen consumption is also a practical approach to alleviate hypoxia and enhance PDT. The O_2_ consumptions of tumors often surpass the O_2_ supply because of the high respiration rate of tumor cells [81]. Hence, to decrease the O_2_ demand for tumor cells, inhibition of mitochondrial respiration is an effective method.

Nitric oxide (NO) is a competitor of O_2_ which can bind to the oxygen-binding site of mitochondria. Therefore, NO can inhibit cell respiration and disturb cell metabolism as an O_2_ economizer [82,83]. Based on this evidence, Yu et al. constructed PLGA nanovesicles to form a PDT-specific O_2_ economizer. These polymeric nanovesicles (PVs) were co-loaded with a NO donor, sodium nitroprusside (SNP), and the photosensitizer TPP (denoted as PV-TS). After the accumulation of this system into the tumors, the released SNP reacted with overexpressed thiol compounds in tumor cells to form S-nitrosothiol. Then, NO could spontaneously be produced by the S-nitrosothiol, inhibiting cellular respiration, and enhancing PDT efficacy. In this study, the oxygen consumption was measured by cytochrome *c* oxidase (C*c*O), an enzyme that converts oxygen to water in the respiration chain. The results showed that the C*c*O activity was reduced with increased PV-TS concentrations, indicating the ability of PV-TS to inhibit cellular respiration and save O_2_ consumption. The research also demonstrated that almost all the PV-TS treated cells died under the irradiation, while about 40% cells remained alive after the treatment of PV-T, showing the superiority of PVTS over PV-T in PDT efficacy due to the presence of SNP [84].

The oxidative phosphorylation (OXPHOS) process expenses a large amount of O_2_ in mitochondria, and the inhibition of OXPHOS may be a promising strategy for mitigating hypoxia [85]. Thus, Xia et al. developed NPs containing atovaquone (Ato) and indocyanine green–bovine serum albumin (ICG-BSA) (denoted as Ato-ICG-GNPs). After accumulation at tumor sites, these nanoparticles were cleaved by matrix metallopeptidase 2 (MMP-2) enzyme, releasing Ato and ICG. The research suggested that the oxygen content in the Ato-ICG-GNPs group dramatically reduced both under normoxia and hypoxia. Therefore, as a strong inhibitor of OXPHOS, Ato could suppress the consumption of oxygen, resulting in a relieved hypoxic state and increased PDT efficacy [86]. In addition to Ato, 3-Bromopyruvate (3BP) has also been reported to block the mitochondrial respiratory chain [87]. Wen et al. designed a PLGA nanoplatform encapsulating with 3BP and loading with the photosensitizer IR780 (designated as 3BP@PLGA-IR780), as shown in Figure 7a. This nanoplatform could accumulate in tumor tissues deeply, and it localized at intracellular mitochondria due to the intrinsic characteristics of IR780. Figure 7b showed that 3BP@PLGA reduced oxygen consumption of tumor cells compared with the control group, while 3BP@PLGA-IR780 indicated better inhibition effects on oxygen consumption. In addition, the research also found that 3BP activated the mitochondrial apoptotic signaling pathway, since apoptotic correlation factors including Cytc and Caspase-3 showed higher expression, as shown in Figure 7c. Hence, 3BP could not only suppress oxygen consumption by inhibition of mitochondrial respiratory chain, but also could disturb the energy metabolism of tumor cells [88].

However, the reduced amount of oxygen consumption by inhibition of respiration also has a limitation. Antitumor agents, such as SNP or 3BP, may also cause damage to normal tissues. Therefore, attention must be paid to the accurate targeting ability of these agents.

## 5. Regulation of Tumor Microenvironments

Solid tumors are composed of cancer cells and tumor microenvironments. The major components of the tumor microenvironment include vasculature, cancer-associated fibroblasts, immune cells, endothelial cells, and an extracellular matrix [89]. Abnormalities in the tumor microenvironment, characterized by lower O_2_ content, lower pH and vascular abnormality, can accelerate tumor progression and treatment resistance [90,91]. Moreover, it has been evidenced that the normalization of the microenvironment can improve treatment outcomes in mice and patients with malignant and nonmalignant diseases [92]. Therefore, many efforts have also been made in the regulation of tumor microenvironments to ameliorate hypoxia, apart from using O_2_ carriers or O_2_ generation materials.

### 5.1. Improving Blood Flow

It is recognized that tumor hypoxia is caused by the abnormal vascular network, which induces decreased blood perfusion and oxygen partial pressure [93,94,95]. Therefore, improving blood flow has become an effective means to increase the amount of O_2_ in tumors. Previous studies have found that increasing local temperatures through mild heating (40–42 °C) leads to an increase in tumor blood flow and elevation of the O_2_ level inside tumors [96,97]. The hyperthermia produced by PTT is the most common approach to increase the temperature, and the pretreatment of tumors with mild hyperthermia could effectively minimize hypoxia, making them more susceptible to PDT [98]. Therefore, the combination of PTT and PDT has attracted significant attention recently. For instance, Feng et al. constructed a NIR light-activatable liposomal Ce6 agent constructed by co-encapsulating a hexylamine conjugated Ce6 (*h*Ce6) and DiR molecules into the bilayers of PEG-shelled liposomes. In this system, the photosensitizing effect of *h*Ce6 was quenched by DiR via fluorescence resonance energy transfer (FRET). Under the 785 nm laser irradiation, however, the fluorescence and photodynamic effect of *h*Ce6 would be activated because of the photobleaching of DiR. Significantly, the 785 nm laser irradiation could lead to mild photothermal heating to enhance the tumor blood flow and efficiently relieve tumor hypoxia, because the ex vivo experiment results showed that the percentage of the hypoxia-positive area considerably dropped from ∼38% to ∼12% after mild NIR-induced photothermal heating with DiR-hCe6-liposome. This strategy showed the potential of the DiR-hCe6-liposome to enhance the effective synergistic therapeutic effect in PDT [99]. Although mild heating effectively enhances the PDT, it can also possibly cause side effects in human bodies when applied in clinical cancer treatment.

In addition, chemotherapeutic drugs, including taxane, gemcitabine, cyclophosphamide, and cisplatin, have also been employed to regulate the disordered microvascular structure of tumors, increasing the blood perfusion and O_2_ concentration [100,101,102]. For example, Shen et al. found that thalidomide (Thal) could stabilize the tumor vessel, resulting in a reduction in tumor vessel tortuosity and leakage, and increased vessel thickness and tumor perfusion. Moreover, the delivery and efficacy of cisplatin were greatly increased through the normalized tumor vasculature, which led to profound antitumor effects [103]. In addition, glucocorticoid steroid dexamethasone (DEX) was found to be able to normalize tumor vessels and the extracellular matrix [104]. Therefore, Zhu et al. prepared a PDT system based on aggregation-induced emission luminogen (AIEgen) called tumor-exocytosed exosome/AIEgen hybrid nanovesicles (DES), shown in Figure 8a. In this system, dexamethasone was used to normalize vascular function in the tumor microenvironment and improve the available O_2_ supply in the tumor. The results of in vivo study in this research showed that the combination of DES with laser irradiation induced the partial inhibition of tumor growth, while the inhibition was significantly more potent when treated with DEX (Figure 8b,c). Figure 8d showed the enhanced ROS production via DES + DEX + L combination treatment. This in vivo research indicated that the combination of DEX with the EXO/AIEgen hybrid nanovesicles facilitated efficient tumor penetration, overcame the hypoxia, and enhanced PDT [105]. However, drugs often have the first pass effect, where drugs are metabolized at a specific location in the body, resulting in a reduced concentration of the active drug at its site of action.

### 5.2. Regulation of HIF-1 Expression

Hypoxia-inducible factor, HIF-1, affects many downstream gene transcriptions, including glucose metabolism, cell proliferation, migration, and angiogenesis. The hypoxic state in tumor cells can activate and induce high expression of HIF-1 [106,107]. Therefore, direct blocking of HIF-1 leads to the change in TME and reprogramming of hypoxic metabolism of tumor cells, thus overcoming hypoxia and enhancing PDT. Based on the intracellular signaling pathways of HIF-1, many research studies have been worked on to precisely block HIF-1α using small molecular inhibitors [108,109,110].

Acriflavine (ACF) is a HIF-1α inhibitor that can inhibit the dimerization of HIF-1α. Cai et al. constructed MOF nanoparticles PCN 224, which were composed of the photosensitizer H_2_TCPP, Zirconium ions, ACF, immunologic adjuvant cytosine–phosphate–guanine (CpG), and surface coating of HA. The designed MOF NPs, PCN-ACF-CpG@HA, could target tumor cells via HA, block the HIF-1 mediated survival and metastasis signaling of ACF and boost strong host anticancer immune responses to eliminate residual cancer cells with CpG, thus enhancing the therapeutic effects of PDT. Notably, since ACF blocked the dimerization of HIF-1α with HIF-1β rather than directly affecting the expression of HIF-1α, the researchers evaluated the downregulation of HIF-1α by detecting the expression of HIF-1α-regulated and survival/metastasis-associated crucial genes. The results showed that these genes were all downregulated under the treatment of PCN-ACF-CpG@HA, indicating that the NPs could considerably inhibit the HIF-1α mediated cell survival and metastasis signaling. An in vitro synergistic antitumor effect study of PCN-ACF-CpG@HA showed that the cell viability of the cells treated with 32 μg mL^−1^ PCN-ACF-CpG@HA under laser irradiation was 11%. In sharp contrast, the cell viability of the cells treated with PCN-ACF-CpG@HA without laser was as high as 75%, clearly demonstrating the dramatically improved antitumor effects of PCN-ACF-CpG@HA by inhibiting HIF-1α survival signaling with ACF [111].

In addition, curcumin (Cur), a natural medicinal ingredient, much evidence has also indicated that Cur can downgrade HIF-1α levels and deplete GSH at the same time [112]. Inspired by this, a PDT-integrated chemotherapy nanoparticle (ZnPc@Cur-S-OA) was developed. Upon 638 nm laser irradiation, the photosensitizer zinc(II) phthalocyanine (ZnPC) could generate singlet oxygen to kill the tumor cells and activate the prodrug Cur-S-OA to release free Cur. Cur-S-OA in the ZnPc@Cur-S-OA NPs was not only able to improve drug loading, but also was able to downregulate GSH and HIF-1α levels to achieve the combination therapy with PDT [113].

Rapamycin (RAP) is an mTOR inhibitor that is identified as an upstream activator of HIF-1 function. Therefore, RAP can inhibit mTOR and thus leads to the blockage of HIF-1α expression and HIF-1-dependent transcription [114]. Liu et al. designed a catalase-loaded MOF which was coated with the nanocore formed by the self-assembly of RAP and Ce6 (denoted as RC@TFC NPs). In this system, catalase functioned as a catalyst for the decomposition of H_2_O_2_ to generate O_2_, and the released RAP acted as an inhibitor for the downregulation of HIF-1α. Western blot analysis showed that RC@TFC displayed ∼70% HIF-1α inhibition and confirmed the successful inhibition of mTOR. In addition, the in vivo experiments showed that without irradiation, the RC@TFC NPs could moderately inhibit tumors due to the chemotherapy of RAP. Combined with laser irradiation, however, the tumor inhibition effect was significantly enhanced, indicating the superior potential of these NPs for tumor therapy. Overall, RC@TFC NPs were able to accumulate into the tumor, mitigate hypoxia, and downregulate HIF-1α, contributing to considerably increased therapeutic effects of PDT [115].

Although all the research above showed the enhanced efficacy of PDT by inhibition of HIF-1, it was challenging to ensure the inhibition would not affect other pathways.

## 6. Summary and Future Perspective

PDT, as a potential tumor therapy, has made dramatic progress over the past few decades due to its low invasiveness, high viability, and great efficiency. Though PDT shows a great number of advantages, hypoxia in tumors tremendously affects the PDT efficacy owing to the oxygen-dependent character of the PDT process. Therefore, many research studies have focused on designing new strategies to overcome hypoxia in recent years. In this review, we have briefly discussed recently developed methods to alleviate the hypoxia limitation from various directions. On the one hand, oxygen carriers such as hemoglobin, perfluorocarbons, and MOFs can capture exogenous oxygen and directly deliver it to tumors. On the other hand, oxygen can be generated endogenously to increase the oxygen supply in the tumor through the decomposition of H_2_O_2_, metal oxides, or water. In addition, reducing the O_2_ consumption through the inhibition of the mitochondrial respiration chain is also an effective strategy. Other strategies to regulate the tumor microenvironment, including improving the blood flow and inhibition of HIF-1, have also been discussed here. These research studies suggest that those approaches are effective when it comes to mitigating hypoxia environments in tumors and improve PDT efficacy.

Though many efforts have been made to mitigate tumor hypoxia, there are still some limitations and challenges to the application of clinical cancer treatment. Some of them have been discussed above. For example, to apply PDT into the clinical treatment, PDT systems must fulfill several requirements, including biosafety, biocompatibility, and biodegradability. For example, when nanosystems containing metal oxides and MOFs are used in PDT, their biosafety and degradation methods must be cautiously assessed due to the harmful impact of metal ions on human bodies. Until now, most inorganic NPs have still been in the preclinical research stage, and more research is needed to evaluate whether they can be utilized safely and effectively in human beings. Additionally, when using agents to inhibit the respiration or HIF-1 expressions, it is imperative to ensure that these agents target correctly to the tumor site and will not affect other normal tissue or pathways.

In the type I mechanism of the ROS generation process in PDT, the electron or hydrogen atom is transferred from the triplet excited state of PS to the molecular oxygen to generate peroxide ions, hydrogen peroxide, and hydrogen radicals. Therefore, the type I mechanism is less oxygen dependent, and thus is less affected by hypoxia compared to the type II mechanism [17]. Some photosensitizers, such as bacteriochlorin, can generate singlet oxygen and radicals through the type I mechanism [116]. Thus, combined effects of both singlet oxygen and radical species for enhanced photodynamic effect may be a promising way to address hypoxia and improve PDT efficacy. Furthermore, the combination of PDT with other therapies could also provide an innovative strategy to relieve hypoxia. For example, PDT efficacy can be enhanced by the combination of PDT with chemotherapy or photothermal therapy [117,118]. Finally, the cellular environments in tumors are very complex, and there are still many unsolved problems and unknown mechanisms. Hence, a further and deeper understanding of the intracellular mechanism inside tumors may provide more promising strategies for effective cancer therapy.

By solving challenges and understanding unknown mechanisms, potent and versatile strategies will open a new path for PDT to encourage its therapeutic chances for clinical cancer treatment in near future.

## Figures and Tables

**Figure 1 pharmaceutics-14-01763-f001:**
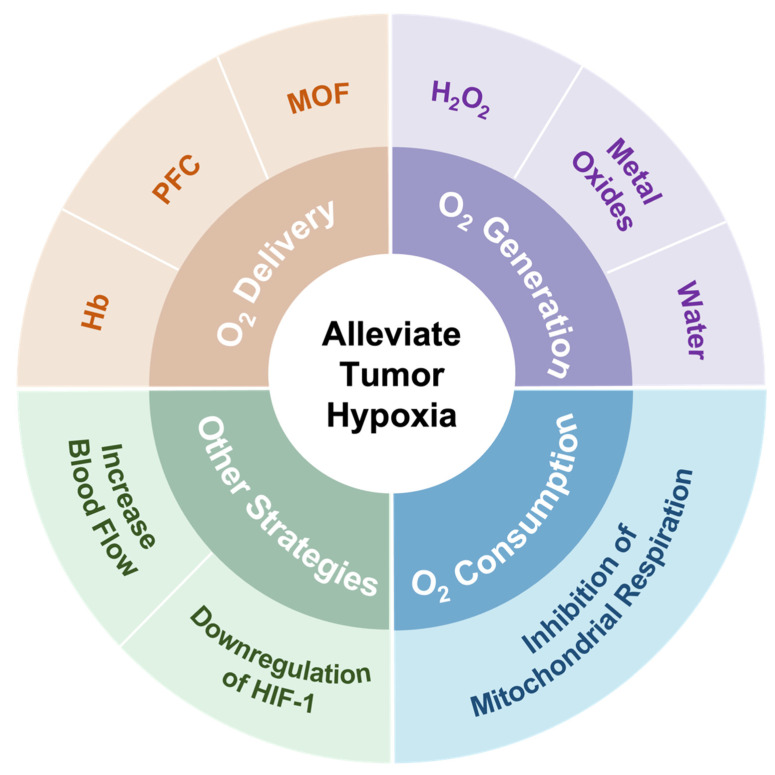
Summary of strategies employed to address hypoxia in the tumor microenvironment.

**Figure 2 pharmaceutics-14-01763-f002:**
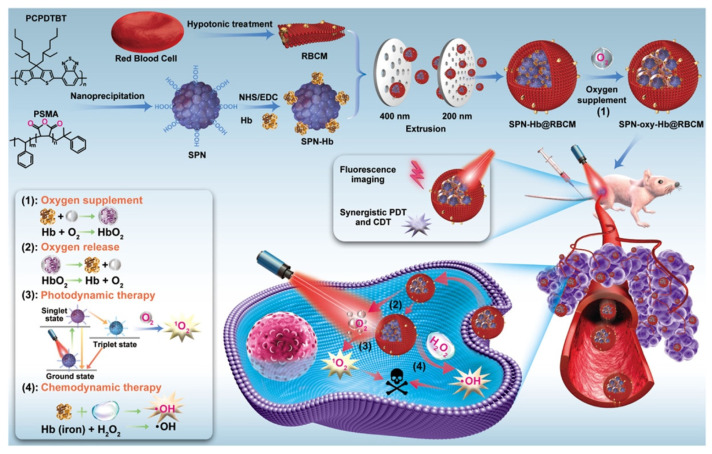
Illustration for the preparation of the SPN-oxy-Hb@RBCM theranostic nanoagent and its application in fluorescence imaging (FI)-guided chemo- and photodynamic therapy of solid hypoxic tumors. Adapted with permission from Ref. [30]. Copyright 2021, American Chemical Society.

**Figure 3 pharmaceutics-14-01763-f003:**
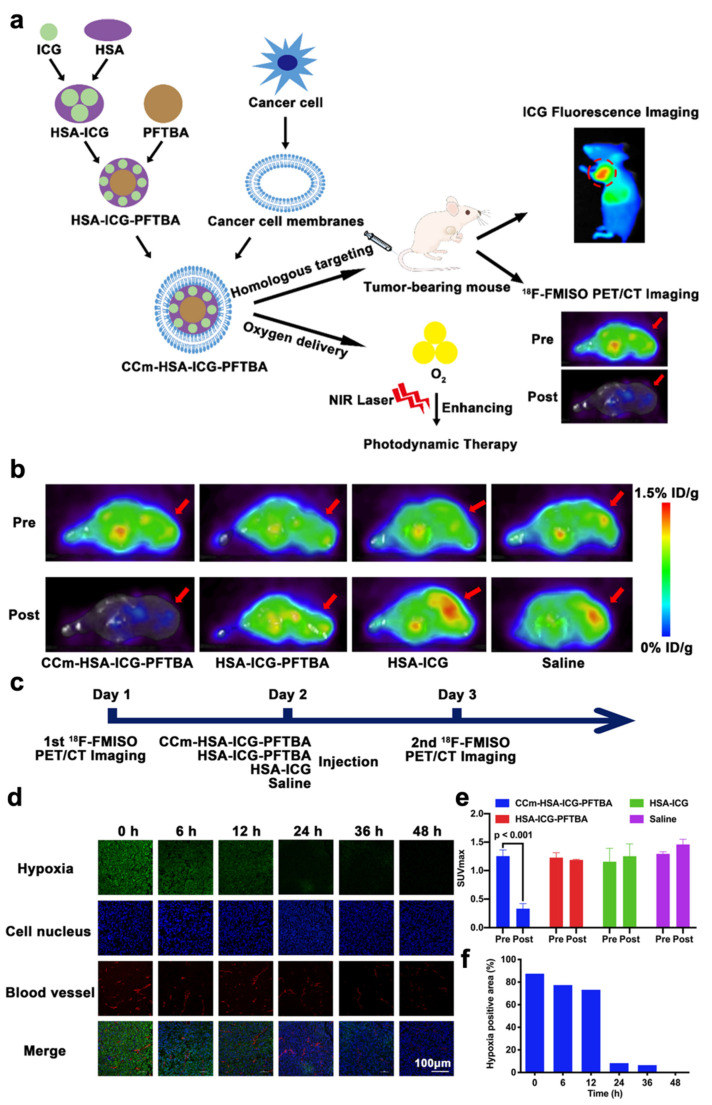
Summary of the design of CCm–HSA–ICG–PFTBA and the results of hypoxia improvement at tumor sites. (**a**) Illustration of CCm–HSA–ICG–PFTBA for homologous targeting and improving oxygen concentration at tumor sites. (**b**) In vivo transverse 18F-FMISO PET/CT images of TNBC xenografts before and after the injection of the CCm–HSA–ICG–PFTBA, HSA–ICG–PFTBA, HSA–ICG, and saline. (**c**) Scheme of the PET/CT imaging. (**d**) Immunofluorescence images of tumor slices stained by the hypoxyprobe. (**e**) The quantitative analysis of SUVmax at tumor sites in the pre and post 18F-FMISO PET/CT imaging. (**f**) Quantification of tumor hypoxia densities for different time points. Adapted with permission from Ref. [34]. Copyright 2021, BioMed Central Ltd.

**Figure 4 pharmaceutics-14-01763-f004:**
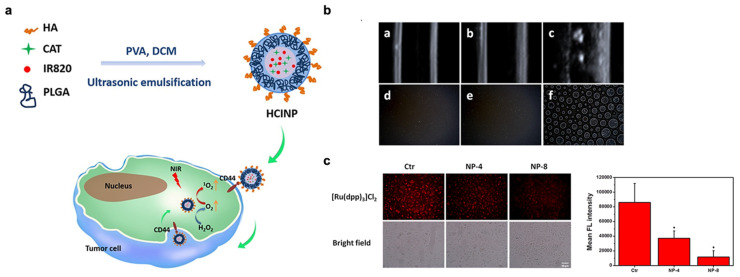
An O_2_-generation strategy by the decomposition of CAT-catalyzed H_2_O_2_ decomposition. (**a**) Schematic illustration of the HCINP assembly and structure, and the mechanism of enhanced PDT efficacy for tumor treatment. (**b**) The result of ultrasound imaging ((**a**) HINPs + H_2_O_2_, (**b**) HCINPs, and (**c**) HCINPs + H_2_O_2_) and optical microscopy ((**d**) HINPs + H_2_O_2_, (**e**) HCINPs, and (**f**) HCINPs + H_2_O_2_) indicating the oxygen generation. (**c**) Intracellular generation of oxygen after HCINPs exposure. * *p* < 0.05, versus the control (Ctr) group. Adapted with permission from Ref. [52]. Copyright 2020, Dove Press Ltd.

**Figure 5 pharmaceutics-14-01763-f005:**
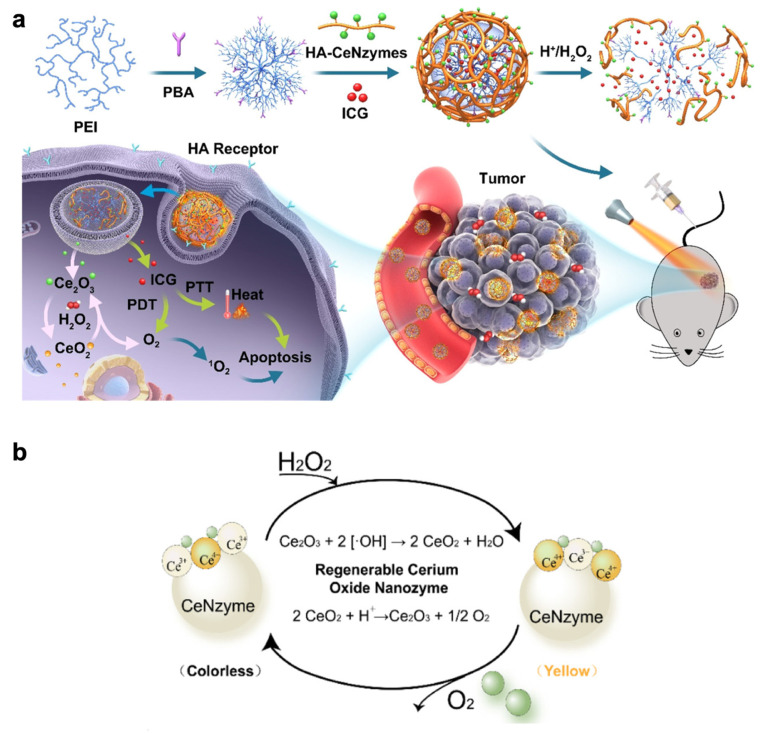
Summary of designed ICG@PEI − PBA − HA/CeO_2_. (**a**) Illustration of nanovesicle, ICG@PEI − PBA − HA/CeO_2_, for tumor-targeted PTT and PDT. (**b**) Schematic indication of the regeneration cycle of cerium oxide. Adapted with permission from Ref. [56]. Copyright 2020, American Chemical Society.

**Figure 6 pharmaceutics-14-01763-f006:**
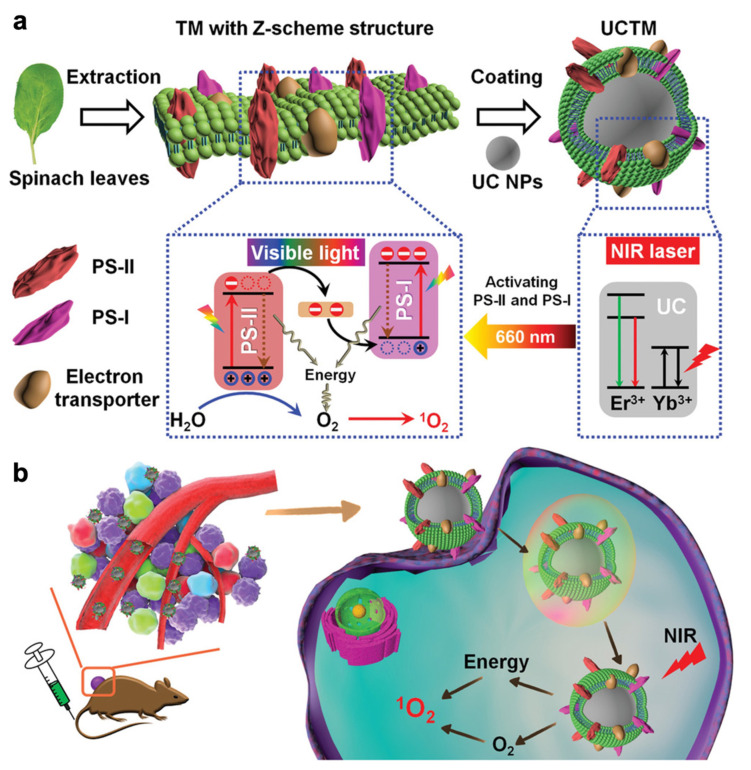
Illustration of photodynamic therapeutic effect against hypoxic tumor of UCTM NPs. (**a**) Preparation Scheme of UCTM NPs, and the electron transfer process as well as O_2−_ and ROS-generation mechanisms of UCTM NPs under NIR laser irradiation; (**b**) the therapeutic process through simultaneously supplying O_2_ and generating ROS under NIR laser irradiation. Adapted with permission from Ref. [66]. Copyright 2021, John Wiley and Sons.

**Figure 7 pharmaceutics-14-01763-f007:**
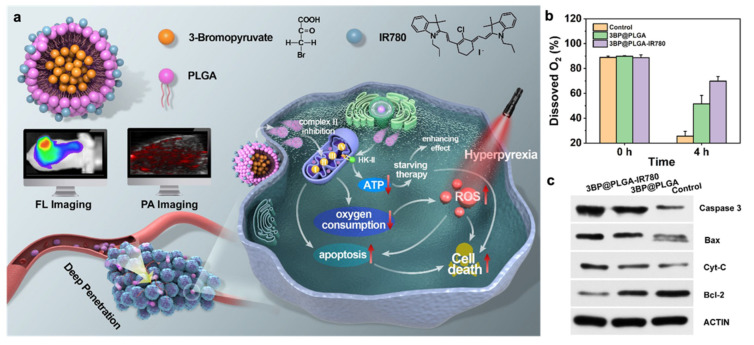
An effective strategy to reduce oxygen consumption by mitochondrial respiration inhibition with 3BP. (**a**) Illustration of designed 3BP@PLGA-IR780. (**b**) The changes of dissolved O_2_ of cell culture medium after incubation with 3BP@PLGA or 3BP@PLGA-IR780 for 4 h. (**c**) The changes of apoptotic proteins in 4T1 cells after various treatments. Adapted with permission from Ref. [88]. Copyright 2021, BioMed Central Ltd.

**Figure 8 pharmaceutics-14-01763-f008:**
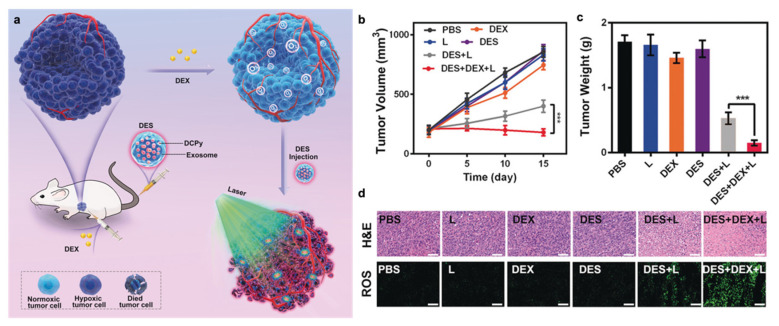
Design of DES nanovesicles and the result of tumor growth inhibition and ROS production. (**a**) Illustration of DES facilitating efficient tumor penetration and PDT. (**b**) Changes in tumor volume over time in response to the indicated treatments. *** *p* < 0.005. (**c**) Average tumor weight values following the indicated treatments. *** *p* < 0.005. (**d**) H&E-stained tumor sections from the indicated treatment groups and ROS measurement by DCFH-DA staining in tumor sections. Adapted with permission from Ref. [105]. Copyright 2020, John Wiley and Sons.

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
