# Peer review of "Recent Strategies to Address Hypoxic Tumor Environments in Photodynamic Therapy"

_pharmaceutics, 2022, doi:10.3390/pharmaceutics14091763_

Round 1
Reviewer 1 Report
The topic of the review article "Recent Strategies to Address Hypoxic Tumor Environments in Photodynamic Therapy" is very relevant and interesting. The authors considered a variety of approaches to overcome tumor hypoxia in order to increase the effectiveness of photodynamic therapy. The work was done at a high level, and deserves to be published in the journal Pharmaceutics after a minor correction. I have no major remarks on the material of the article.
As minor remarks, I will point out the following:
1. It would be appropriate in each chapter to indicate the possible limitations and disadvantages of using these methods. Some of them are presented in the conclusion, but not all.
2. It would be convenient for the reader if for each case, where possible, specific values for the increase in the saturation of tumor cells or tissues (or another indicator) would be given. For example, in sentences with references to sources 23, 24, 32, 47, 57 and 77.
Reviewer 2 Report
The work is described very well. The figures have high quality.
1. In the introduction one part about materials, which are used as fotosensitizers and which could enhance the hypoxic should be added e.g.: 10.1039/c7nr04881e; https://doi.org/10.1186/s12951-019-0539-2; https://doi.org/10.1016/j.pdpdt.2019.101594; https://doi.org/10.1007/s10853-019-04187-z; 10.1080/21691401.2017.1420072; https://doi.org/10.1016/j.physe.2016.03.023
Reviewer 3 Report
I read the review article titled "Recent Strategies to Address Hypoxic Tumor Environments in Photodynamic Therapy" with great interest. The article is well written and comprehensive collection of strategies for alleviating tumor hypoxia. The article can be considered for publication after minor revision. Authors should address the following to improve the article:
1. Authors need to provide a figure summarizing briefly all the strategies employed to address hypoxia in the tumor microenvironment, as Figure 1.
2. " Prussian Blue nanoparticles" is one of the compounds decomposing endogenous H2O2 and aiding in PDT. Authors need to include studies where prussian blue was used as carrier to photosensitzers and potentiated PDT in vitro and in vivo.
Reviewer 4 Report
Du and co-worker present in their submission to Pharmaceutics "Recent Strategies to Address Hypoxic Tumor Environments in Photodynamic Therapy". The main deficit of the review is the uncritical attitude towards the published results in the literature (see below). The English of the submission must be improved, some selected examples are mentioned below. The revised version must be checked again.
Improve the language of the following part: "Additionally, evidence has also proved that"
"Under the laser irradiation, the PS would be activated" should be "Under the laser irradiation, the PS is activated"
"Four molecular O2 can bind to the heme group strongly and can be carried from the lungs to other tissues " should be "Each of the four heme groups can strongly bind to one molecular O2 molecule, which can be carried from the lungs to other tissues "
"the overall oxygen-carrying capacity of SPN-Hb@RBCM nanoparticles was determined and calculated to be 0.142 μg of O2 per 1 mg NPs, confirming the excellent oxygen-carrying capacity of this nanosystem." The oxygen-carrying capacity is only 0.0142 m/m %! It is highly doubtful, whether this is really "excellent". More relevant is that the ratio of PS to transported O2 is not given. The review must mention that this is an essential parameter, which must be mentioned in future O2-carrying PDT publications.
"The in vitro study revealed that at standard atmospheric pressure and room temperature, CuTz-1@F127 could adsorb up to 400 μmol g−1 of O2." The authors should additionally mention how much O2 in mass this corresponds (in order to be able to compare this number with the one mentioned in the previous paragraph).
"Water can generate O2 by using materials such as C3N4 or through photosynthesis." Should be "C3N4 or photosynthesis can turn water into O2."
The PTT is not explained, when it is first mentioned in the manuscript text.
"Since H2O2 is reported to be at high concentration in tumor tissues, H2O2 is the most common oxygen source as it can decompose to generate O2." "The amount of endogenous H2O2 in the tumor is limited to 10-50 μM, which is not enough for the in situ generation of oxygen" These two statements contradict each other.
"Chlorophyl" typo.
"4h+" must be "4H+"
"Finally, the cellular environments in tumors are very complexed" should be "Finally, the cellular environments in tumors are very complex"
"For instance, the molecular structure that mediates important functions of HIF-1 has not yet been well-understood.[110]" This sentence must be re-written as ref. 110 does not talk about the structure of HIF-1.
Ref. 9: For articles with article numbers instead of consecutive page numbers, the article number should be given (here: e101064). Applies also to refs. 26, 28, 33, 35, 52, 53, 56, 59, 71, 77, 79, 94, 102 and 110.
Ref. 9: The first author should be written like this: Kilic Eren, M.
Refs. 20, 61, 69: The article number are missing.
Ref. 41: The page numbers are wrong, they need to be replaced by the article number.
Ref. 84: Several first names are not abbreviated.
Ref. 88: "Tumour PO2 Can Be Increased Markedly by Mild Hyperthermia" should be "Tumour pO2 Can Be Increased Markedly by Mild Hyperthermia"
Ref. 95: volume and page numbers are missing.
Round 2
Reviewer 2 Report
I accept revised version of this manuscript.
Author Response
Thank you so much for your acceptance.
Reviewer 4 Report
Du and co-worker have improved their submission to Pharmaceutics "Recent Strategies to Address Hypoxic Tumor Environments in Photodynamic Therapy". The two remaining issues must be fixed and then publication can proceed.
Equation 3 is incorrect and should be as follows: 2 H2O -> O2 + 4 H+ + 4e-
Ref. 9: The first author should still be written like this: Kilic Eren, M.
Author Response
We are appreciated receiving your valuable and instructive comments. We have studied comments carefully and have corrected the revised manuscript, which we hope to meet with your approval.
Point 1: Equation 3 is incorrect and should be as follows: 2 H2O -> O2 + 4 H+ + 4e-
Response 1: Thank you for your kind suggestions. It is our carelessness to make this mistake in the manuscript. We have corrected it in the manuscript carefully in Page 20 Line 7.
Point 2: Ref. 9: The first author should still be written like this: Kilic Eren, M.
Response 2: It is our carelessness to make this mistake in the manuscript. We have corrected it in the manuscript Page 32 Line 16 (Ref. 9).